# Corticosteroids for severe acute exacerbations of chronic obstructive pulmonary disease in intensive care: From the French OUTCOMEREA cohort

**Louis Marie Galerneau**[1,2]*, **Sébastien Bailly**[2], **Nicolas Terzi**[1,2], **Stéphane Ruckly**[3], **Maïté Garrouste-Orgeas**[4], **Yves Cohen**[5], **Vivien Hong Tuan Ha**[6], **Marc Gainnier**[7], **Shidasp Siami**[8], **Claire Dupuis**[9], **Michael Darmon**[10], **Elie Azoulay**[10], **Jean-Marie Forel**[11], **Florian Sigaud**[1], **Christophe Adrie**[12], **Dany Goldgran-Toledano**[13], **Alexis Ferré**[14], **Etienne de Montmollin**[15,16], **Laurent Argaud**[17], **Jean Reignier**[18], **Jean-Louis Pepin**[2], **Jean-François Timsit**[15,16], on behalf of the OUTCOMEREA network¶

**1** Medical Intensive Care Unit, University Hospital of Grenoble Alpes, Grenoble, France, **2** HP2 laboratory, Grenoble Alpes University, INSERM U1300, Grenoble, France, **3** Department of Biostatistics, Outcomerea, Paris, France, **4** French and British Institute, Medical Unit, Levallois-Perret, France, **5** Intensive Care Unit, Avicenne Hospital, AP-HP, Paris, France, **6** Medical Intensive Care Unit, Meaux hospital, Meaux, France, **7** Medical Intensive Care Unit, La Timone Hospital, Marseille, France, **8** Critical Care Medicine Unit, Etampes-Dourdan Hospital, Etampes, France, **9** Medical Intensive Care Unit, University Hospital of Clermont-Ferrand, Clermont-Ferrand, France, **10** Intensive Care Unit, Saint-Louis Hospital, AP-HP, Paris, France, **11** Medical Intensive Care Unit, Nord University Hospital, Marseille, France, **12** Polyvalent Intensive Care Unit, Delafontaine Hospital, Saint-Denis, France, **13** Medical Intensive Care Unit, Le Raincy-Montfermeil Hospital, Montfermeil, France, **14** Intensive Care Unit, Versailles Hospital, Le Chesnay, France, **15** Medical and Infectious diseases Intensive Care Unit (MI2), Bichat Hospital, AP-HP, Paris, France, **16** IAME, University of Paris, INSERM U1137, University of Paris, F-75018, Paris, France, **17** Medical Intensive Care Unit, Edouard Herriot Hospital, Lyon Civil Hospices, Lyon, France, **18** Medical Intensive Care Unit, Nantes University Hospital, Nantes, France

¶ The members of the OUTCOMEREA study group are listed in the supporting information
* lmgalerneau@chu-grenoble.fr

**Data Availability Statement:** All relevant data are within the manuscript and its Supporting Information files.

## Abstract

### Introduction

Acute exacerbation of chronic obstructive pulmonary disease (COPD) is a frequent cause of intensive care unit (ICU) admission. However, data are scarce and conflicting regarding the impact of systemic corticosteroid treatment in critically ill patients with acute exacerbation of COPD. The aim of the study was to assess the impact of systemic corticosteroids on the occurrence of death or need for continuous invasive mechanical ventilation at day 28 after ICU admission.

### Methods

In the OutcomeRea™ prospective French national ICU database, we assessed the impact of corticosteroids at admission (daily dose $\geq$ 0.5 mg/kg of prednisone or equivalent during the first 24 hours ICU stay) on a composite outcome (death or invasive mechanical ventilation) using an inverse probability treatment weighting.

**Funding:** The authors received no specific funding for this work.

**Competing interests:** NT is supported by Pfizer for attending meetings and/or travel. This does not alter our adherence to PLOS ONE policies on sharing data and materials. The other authors declare that they have no competing interests.

## Results

Between January 1, 1997 and December 31, 2018, 391 out of 1,247 patients with acute exacerbations of COPDs received corticosteroids at ICU admission. Corticosteroids improved the main composite endpoint (OR = 0.70 [0.49; 0.99], p = 0.044. However, for the subgroup of most severe COPD patients, this did not occur (OR = 1.12 [0.53; 2.36], p = 0.770). There was no significant impact of corticosteroids on rates of non-invasive ventilation failure, length of ICU or hospital stay, mortality or on the duration of mechanical ventilation. Patients on corticosteroids had the same prevalence of nosocomial infections as those without corticosteroids, but more glycaemic disorders.

## Conclusion

Using systemic corticosteroids for acute exacerbation of COPD at ICU admission had a positive effect on a composite outcome defined by death or need for invasive mechanical ventilation at day 28.

## Introduction

Severe acute exacerbation of chronic obstructive pulmonary disease (AECOPD) is a frequent cause of admission to an intensive care unit (ICU) and may require non-invasive or invasive ventilation support [1].

Systemic corticosteroids for COPD exacerbations are recommended by the 2022 GOLD (Global Initiative for Chronic Obstructive Lung Disease) Guideline [2]. This recommendation is based on data from non-ICU populations demonstrating the capability of systemic corticosteroids to shorten recovery time and improve lung function [2]. It has also been demonstrated that lower corticosteroid doses are non-inferior to higher doses, which are associated with an increase in adverse effects [3,4]. The French guidelines do not recommend this treatment for all patients hospitalized for AECOPD but rather prefer a case-by-case decision for use, without specificities for ICU patients [5].

The use of corticosteroids in critically ill COPD patients is poorly documented regarding the phenotype of patients treated, dose, duration and side effects. The two randomized studies addressing corticosteroid efficacy for severe AECOPD in the ICU context describe conflicting results [6,7]. Alia *et al.* reported a shorter duration of non-invasive ventilation (NIV), a shorter length of ICU stay, and less failure of NIV in patients receiving corticosteroids. However, the rate of NIV failure in the placebo group was surprisingly high [6]. Conversely, Abroug *et al.* not found no effect of corticosteroids on mortality, NIV failure, duration of mechanical ventilation or length of ICU stay [7]. These two studies did not reach the expected sample size and do not allow us to conclude about corticosteroids benefits. Moreover, patients with pneumonia were excluded from the two studies, despite the fact that respiratory infectious diseases are the most frequent triggers of AECOPD [8].

In summary, the indications for corticosteroid therapy are currently poorly defined for patients with a severe exacerbation of COPD admitted to an ICU. This is a question of major importance owing the potential high impact in terms of medical and/or economic benefits.

The primary aim of our study was to assess the effects of corticosteroids at admission on death and invasive mechanical ventilation at day 28 after ICU admission in for AECOPD using an inverse probability weighted generalized linear model to control for confounding

factors in an observational prospective database. Secondary objectives were the impact of corticosteroid therapy on length of UCU and hospital stays, duration of ventilation, rates of NIV failure, antibiotics use and the safety of corticosteroids. To address this question, we benefited from the prospective longitudinal database OutcomeRea[TM] accumulating data from 32 French ICUs.

## Materials and methods

### Study design and study population

In this observational prospective cohort study all patients were participants in the OutcomeRea[TM] prospective national database (details of the OutcomeRea[TM] database are presented in the online data supplement). We included adults admitted to one of the 32 ICUs participating in OutcomeRea[TM] with a diagnosis of acute exacerbation of COPD. The diagnosis of EACOPD was define by a main diagnosis of *exacerbation of COPD* registered in the database or by a main diagnosis registered of *acute respiratory failure* with a medical history of *COPD* registered in the database. If there was a strong clinical suspicion of COPD, the ICU practitioner could register the presence of COPD in the database. This situation is frequent in daily practice. Indeed, a large part of patients admitted in ICU for severe acute respiratory failure, especially with hypercapnic respiratory failure, have an undiagnosed COPD [9]. Patients included was admitted in ICU between January 1, 1997 and December, 31 2018. This study did not require individual patient consent because it involved research on a previously approved database by our institutional review board of Clermont-Ferrand, France (IRB no. 5891; Ref: 2007–16), which waived the need for signed informed consent of the participants, in accordance with French legislation on non-interventional studies. The exclusion criteria are summarized in Fig 1.

### Methods and measurements

The prescription of corticosteroid therapy is a daily dose $\geq$ 0.5 mg/kg of prednisone (or equivalent) during the first 24 hours after admission to the ICU. Data on patient characteristics, clinical severity scores, biological data, status at the end of the ICU stay, prescription of antibiotics and any type of ventilatory support were extracted from the database. Patients were classified into one of three groups. Data on '*Very Severe COPD*', defined as home oxygen therapy or NIV, or a Forced Expiratory Volume in 1 second < 30% predicted value (GOLD classification, stage 4). Patients with no confirmed mention of these criteria were classified as presenting '*Not very severe COPD*'. Patients for whom COPD was poorly monitored, without recent spirometry or for whom data were not available were classified as '*Unknown COPD severity*'.

The primary endpoint was a composite outcome including death or invasive mechanical ventilation (IMV) at day 28 after admission to the ICU. We compared patients who received corticosteroid therapy with those who didn't. Subgroup analyses were performed on the primary endpoint according to the severity of COPD, type of ventilatory support at admission (invasive or non-invasive) and severity score at admission.

The secondary endpoints were: survival at D28 and D90, NIV failure (defined as death under NIV or necessity for IMV for patients treated by NIV in first intention), length of stay (LOS) in ICU and in hospital, ventilator-free days at D28 for invasive mechanical ventilation, ventilator-free days at D-28 for non-invasive ventilation, prescription of antibiotics and antibiotic-free days at D5, D10, and D15.

We also compared the prevalence of corticosteroids side effects between the two groups (see online data supplement for details about side effects definitions).

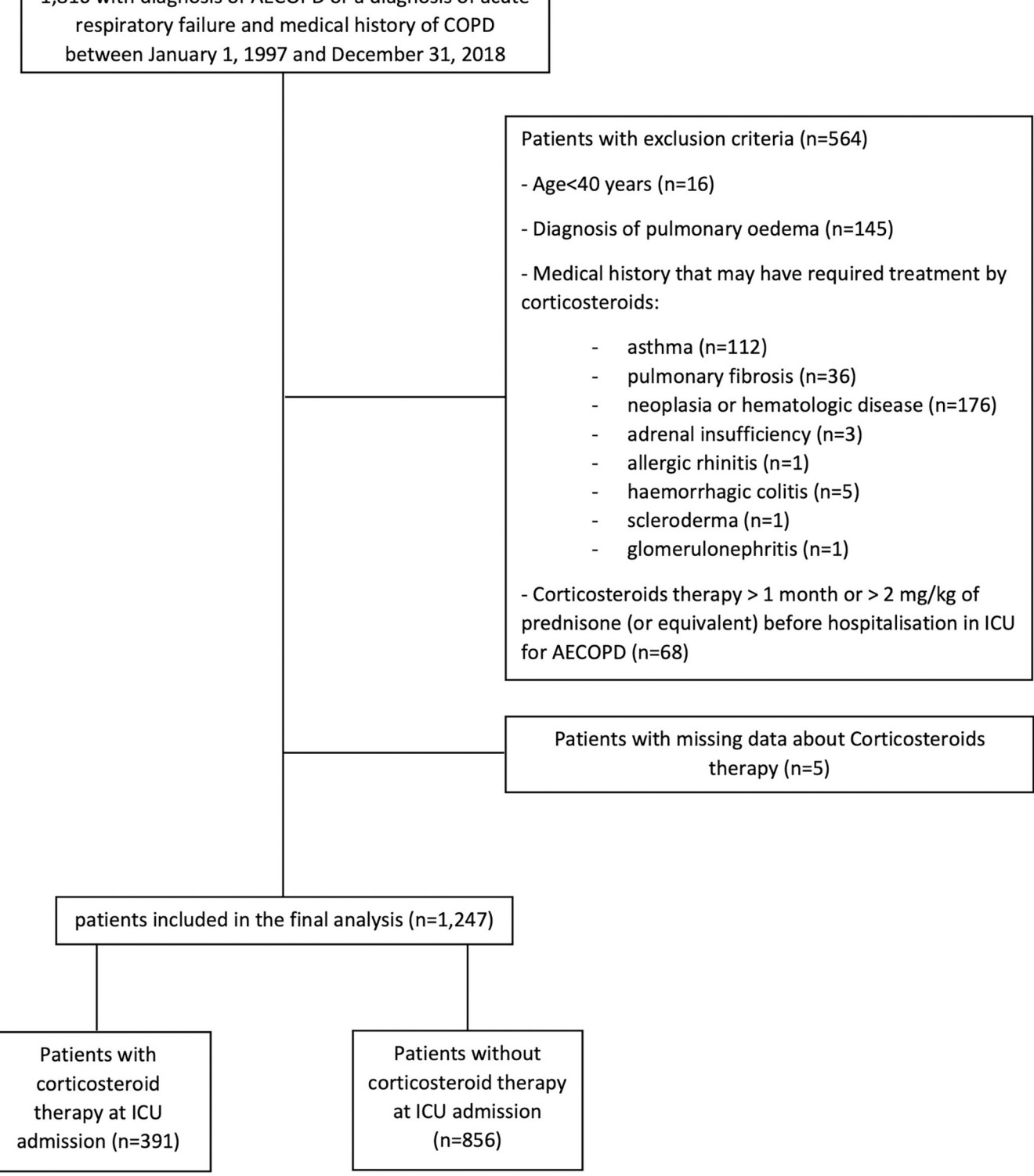

**Fig 1. Study flow chart.** AECOPD: Acute exacerbation of chronic obstructive pulmonary disease. COPD: Chronic obstructive pulmonary disease.

## Statistical analysis plan

A descriptive analysis of the population is presented using frequencies and percent for qualitative data and median and interquartile range (IQR) for quantitative data.

To estimate the average causal effect of corticosteroids at admission on endpoints, an inverse probability of treatment weight (IPTW) estimator was used (see online supplements) [10,11]. Logistic regression, using the IPTW, was used to assess the impact of corticosteroids at admission on the composite outcome. Logistic regression, Cox models or negative binomial regressions using the IPTW were used for secondary outcomes. Comparisons of adverse events between groups were performed with Chi2 test and Wilcoxon-Mann-Whitney test (see (see online data supplement for details).

A p value less than 0.05 was considered significant. Statistical analyses were performed using SAS 9.4 software© (SAS Institute, Cary, NC, USA).

## Results

### Characteristics of the study population

Among the 23,249 patients registered in the OutcomeRea[TM] database between January 1, 1997 and December 31, 2018, 1,816 patients were recorded as having a diagnosis of severe AECOPD, or a diagnosis of acute respiratory failure and a medical history of COPD. After the removal from the study database of 569 patients owing to exclusion criteria, 1,247 patients were included in the final analysis. (Fig 1)

The clinical characteristics of the study population at ICU admission are shown in Table 1. The characteristics of COPD exacerbations and arterial blood gas result which is a prognosis

**Table 1. Baseline characteristics of the population at admission to ICUs for severe AECOPD.**

|  | Median [Q1; Q3] or number (Percentage) |
|---|---|
| **Characteristics and Severity at ICU admission, n = 1,247** | |
| Age (years) | 70 [62; 78] |
| Male sex, n (%) | 803 (64.4) |
| BMI, kg/m$^2$ | 24.9 [20.9; 30.3] |
| SAPS II score | 37.0 [28.0; 48.0] |
| Glasgow Score Scale at admission | 15 [12; 15] |
| PaO$_2$/FiO$_2$ at admission (mmHg) (n = 1173) | 207 [138; 287] |
| <100 | 159 (13.6) |
| 100–199 | 397 (33.8) |
| 200–299 | 347 (29.6) |
| ≥ 300 | 270 (23.0) |
| SOFA Score Day-1 | 4 [2; 6] |
| SOFA Score Day-2 | 4 [2; 6] |
| Extra-pulmonary SOFA Score Day-1 | 2 [0; 4] |
| Extra-pulmonary SOFA Score Day-2 | 1 [0; 4] |
| **Comorbid condition** | |
| Arterial hypertension, n(%) | 309 (24.8) |
| Chronic heart disease, n (%) | 255 (20.4) |
| Diabetes mellitus, n (%) | 235 (18.8) |

ICU: Intensive Care Unit. AECOPD: Acute exacerbation of chronic obstructive pulmonary disease. BMI: Body Mass Index. SAPS II: The Simplified Acute Physiology Score II. Pa02: Partial pressure of oxygen. FiO2: Fraction of inspired oxygen. SOFA: Sequential Organ Failure Assessment.

factor [12] are given in Table 2. Very severe COPD was recognized in 265 (21.2%) of the patients, 446 (35.8%) had a non-severe COPD and the COPD severity status was unknown for 536 (43.0%) patients. The main cause of AECOPD was a respiratory infection (893 patients, 72.4%), followed by another respiratory cause (226 patients, 18.3%).

**Table 2. Characteristics of past medical history and COPD exacerbation.**

| | Median [Q1; Q3] or number (Percentage) |
|---|---|
| **COPD characteristics (n = 711)** | |
| Long-term oxygen therapy at home | 220 (30.9) |
| NIV at home | 85 (11.9) |
| Pulmonary function tests (n = 290) | |
| *Stage 1 ($FEV_1 \geq 80\%$ predicted)* | 4 (1.4) |
| *Stage 2 ($50\% \leq FEV_1 < 80\%$ predicted)* | 72 (24.8) |
| *Stage 3 ($30\% \leq FEV_1 < 50\%$ predicted)* | 133 (45.9) |
| *Stage 4 ($FEV_1 < 30\%$ predicted)* | 81 (27.9) |
| **COPD Severity (n = 1,247)** | |
| Very Severe COPD (Oxygen therapy at home or NIV at home or airflow limitation GOLD Stage 4) | 265 (21.2) |
| Not very severe COPD | 446 (35.8) |
| Unknown COPD severity | 536 (43.0) |
| **Cause of acute exacerbation of COPD (n = 1,247)** | |
| Respiratory infection, n (%) | 893 (72.4) |
| Respiratory except respiratory infection, n (%) | 226 (18.3) |
| Cardiac except pulmonary oedema, n (%) | 23 (1.9) |
| Venous thromboembolism, n (%) | 15 (1.2) |
| Postoperative, n (%) | 10 (0.8) |
| Neurologic, n (%) | 19 (1.5) |
| Digestive, n (%) | 15 (1.2) |
| Other, n (%) | 46 (3.7) |
| **Timing to ICU admission (n = 1,247)** | |
| Direct ICU admission or < 24h after hospital admission | 1030 (82.6) |
| ICU admission > 24h and $\leq$ 7 days after hospital admission | 115 (9.2) |
| ICU admission > 7 days after hospital admission | 102 (8.2) |
| **Arterial blood gas at ICU admission** | |
| pH (mmHg) (n = 1108) | 7.33[7.25; 7.40] |
| *< 7.25* | 268 (24.19) |
| *7.25–7.29* | 162 (14.62) |
| *7.30–7.34* | 212 (19.13) |
| *$\geq$ 7.35* | 466 (42.06) |
| $PaO_2$ (mmHg) (n = 1199) | 76 [62; 105] |
| $PaCO_2$ (mmHg) (n = 1200) | 59 [45; 74] |
| $HCO_3^-$ (mmol/L) (n = 1124) | 29 [25; 34] |
| **Limitation of therapeutic effort** | |
| Limitation of therapeutic effort during ICU stay, n (%) | 207 (16.6) |
| Limitation of therapeutic effort at admission in ICU, n (%) | 96 (7.7) |

Stages of airflow limitation defined according to GOLD Report 2022. [2].

COPD: Chronic obstructive pulmonary disease ICU: Intensive Care Unit. $FEV_1$: Forced Expiratory Volume. NIV: Non-Invasive Ventilation. Pa02: Partial pressure of oxygen. FiO2: Fraction of inspired oxygen. PaCO2: Partial Pressure of Carbon Dioxide.

Among the whole study population 391 (31.4%) patients received corticosteroid therapy at ICU admission (initiated during the first 24 hours after ICU admission) and 515 (41.3%) received corticosteroid therapy at least once during their ICU stay. 879 (70.5%) patients were being treated by antibiotics at admission and 974 (78.1%) received antibiotics at least once during the ICU stay.

Most patients (990, 79.4%) required ventilatory support at admission (36.5% IMV and 42.9% NIV) (S1 Table). Among the 625 patients who had NIV as first ventilatory support, 152 (24.3%) experienced NIV failure occurring at a median of 2.0 [2.0; 3.0] days after NIV initiation and 2.5 [2.0; 4.0] days after admission to the ICU.

Characteristics and differences between the two groups (with or without corticosteroids) at admission are shown in the online data supplement (S2 Table).

The major part of patient is directly admitted in ICU or during the 24 hours after hospital admission (1030, 82,6%). The median ICU length of stay was 6 [4; 12] days, and median hospital length of stay was 18 [11; 31] days. The overall mortality was 238 (19.1%) patients, with 151 (12.1%) deaths occurring in ICUs. The 28-day mortality was 14.8% and the 90-day mortality was 27.8% among the 863 patients for whom data were available.

## Primary outcome

In a multivariable analysis, after IPTW weighting (double robust analysis) (weight model used resumed in S3 Table), corticosteroid administration at ICU admission in ICU significantly improved the principal composite outcome (OR = 0.70 [0.49; 0.99], *p = 0.044*) (S4 Table, Fig 2).

The subgroup analyses on the principal composite endpoint are summarized in Fig 2. For the subgroup of patients with very severe COPD, the protective effect of corticosteroid therapy on the primary composite outcome was lost (OR = 1.12 [0.53; 2.36], *p = 0. 770)*.

## Secondary outcomes

Results concerning in-ICU mortality, in-hospital death and survival analysis at 28-days and 90-days are summarized in S5 and S6 Tables and S1–S4 Figs in the online data supplement, respectively.

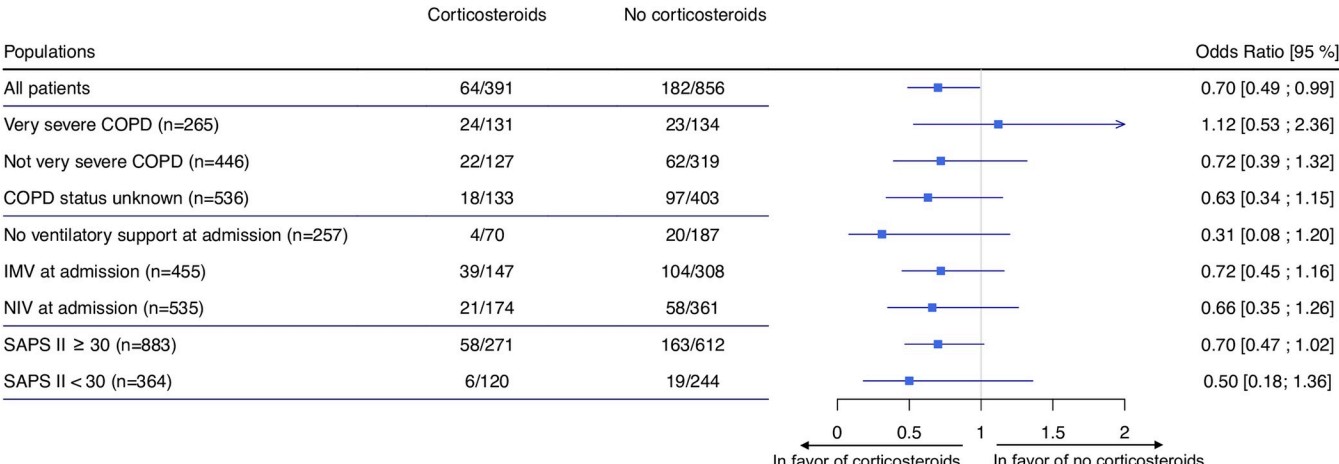

**Fig 2. Summary of results for the primary outcome (death or invasive mechanical ventilation at day 28).** Heterogeneity for COPD severity subgroups: Chi$^2$ = 3.50, df = 2, p = 0.32. Heterogeneity for ventilatory support subgroups: Chi$^2$ = 6.36, df = 2, p = 0.10. Heterogeneity for SAPS II subgroups: Chi$^2$ = 5,26, df = 1, p = 0.15. *COPD*: *Chronic obstructive pulmonary disease. NIV*: *Non-Invasive Ventilation. IMV*: *Invasive Mechanical Ventilation. SAPS II*: *The Simplified Acute Physiology Score II.*

There was no significant impact of corticosteroids on NIV failure rates for patients with NIV ventilatory support as first intention (OR = 0.92 [0.61; 1.41], *p = 0.714*), length of stay in ICU for the 1096 patients alive at ICU discharge from (IRR = 0.93 [0.85; 1.02], *p = 0.111*), and length of hospital stay after ICU admission for the 950 patients alive at ICU discharge with a known length of stay (IRR = 1.00 [0.92; 1.09], *p = 0.946*).

Concerning the duration of ventilation, there was no significant effect of corticosteroids on ventilator-free days (VFD) at 28-days with ventilation corresponding as NIV or IMV in the whole study population (IRR = 1.06 [0.95; 1.19], *p = 0.278*), or for the 1034 patients with at least one day of NIV or IMV during the first 28-days of ICU stay (IRR = 1.04 [0.90; 1.19], *p = 0.606*). There was also no significant effect of corticosteroids on VFD at 28-days with ventilation corresponding only to IMV in the total population (IRR = 1.06 [0.95; 1.19], *p = 0.290*), or for the 540 patients with IMV during the first 28-days of ICU stay (IRR = 1.03 [0.79; 1.36], *p = 0.812*) (S5 Fig).

We found no statistical relation between the prescription of corticosteroid at admission and the prescription of antibiotics during the ICU stay (OR = 1.13 [0.82; 1.56], *p = 0.459*) or prescription of antibiotics at admission to the ICU (OR = 1.20 [0.90; 1.60], *p = 0.203*) (S6 Fig). There was a significant inverse correlation between corticosteroid therapy and antibiotic-free days (AFD) at D10 for the 395 patients with a length of ICU stay ICU $\geq$ 10 days (IRR = 0. 77 [0.59; 0.99], *p = 0.046*). This correlation was not found for antibiotic-free days at D5 for patients with an ICU stay $\geq$ 5 days (n = 824) or for antibiotic-free days at D15 for patients with remaining $\geq$ 15 days in the ICU (n = 242) (S7 Fig).

### Adverse events and corticosteroids (Table 3)

We found no difference regarding the prevalence of nosocomial infectious events between groups (with or without corticosteroids) at admission to the ICU. The maximum systolic blood pressure was significantly higher for patients receiving corticosteroids at admission. Concerning metabolic disorders, the urea ratio (defined by the ratio of maximum urea level during ICU stay to the urea level at admission in mmol/L) was higher for patients treated with corticosteroids. More glycaemic disorder-type events (hyperglycaemia and hypoglycaemia) were experienced by patients treated by corticosteroids. Digestive bleedings were not significantly different between the two groups but gastric protective agents were more often prescribed for the patients with corticosteroid therapy.

## Discussion

In this cohort of 1,247 patients with an AECOPD, 391 (31.4%) patients received corticosteroid therapy at admission to the ICU. Corticosteroids at admission in ICU improved the principal composite endpoint defined as death or invasive mechanical ventilation at Day 28 (OR = 0.70 [0.49; 0.99], *p = 0.044*). Treatment with corticosteroids had no impact on mortality, NIV failures, lengths of stay, or lengths of ventilation. There was a trend for an increased consumption of antibiotics at D10 for patients receiving corticosteroid therapy. No increased prevalence of nosocomial infectious events was found among patients treated with corticosteroids at admission but they more frequently exhibited glycaemic disorders.

The characteristics of our population including age, sex, BMI and COPD severity scores at ICU admission, or outcomes such as lengths of stay or overall mortality of 19.1%, with a 12.1% mortality in the ICU are similar to those of previous studies of AECOPD in the ICU [13–15]. A respiratory infection as the main trigger of AECOPD is also consistent with previous findings [2]. Thus, the generalisability of our results seems good. Moreover, our study by included data from patients admitted to one or other of the 32 ICUs throughout France.

**Table 3. Adverse events according to corticosteroids prescription for AECOPD at admission to ICU.**

| | No Corticosteroids at admission to ICU (n = 856) | Corticosteroids at admission to ICU (n = 391) | p-value |
|---|---|---|---|
| | Median [Q1; Q3] or number (Percentage) | Median [Q1; Q3] or number (Percentage) | |
| **Nosocomial infectious events** | | | |
| At least one nosocomial infectious event, n (%) | 166 (19.4) | 77 (19.7) | 0.901 |
| Bacteraemia, n (%) | 35 (4.1) | 10 (2.6) | 0.179 |
| Nosocomial Pneumonia, n (%) | 78 (9.1) | 45 (11.5) | 0.188 |
| Ventilator-associated pneumonia among 549 patients receiving IMV in ICU | 69 (18.6) | 37 (22.0) | 0.346 |
| Catheter infection, n (%) | 46 (5.4) | 19 (4.9) | 0.704 |
| Urinary tract infection, n (%) | 59 (6.9) | 26 (6.7) | 0.874 |
| Cholecystitis, n (%) | 3 (0.4) | 1 (0.3) | 0.784 |
| Tracheobronchitis, n (%) | 27 (3.2) | 8 (2.1) | 0.272 |
| Viral infection, n (%) | 15 (1.8) | 4 (1.0) | 0.329 |
| Other nosocomial infectious events, n (%) | 8 (0.9) | 7 (1.8) | 0.198 |
| Nosocomial infectious events (first 5days) for patients with LOS in ICU ≥ 5 days (n = 824), n (%) | 56 (10.1) | 18 (6.6) | 0.100 |
| Nosocomial infectious events (10 first days) for patients with LOS in ICU ≥ 10 days (n = 395), n (%) | 85 (31.6) | 34 (27.0) | 0.351 |
| Nosocomial infectious events (15 first days) for patients with LOS in ICU ≥ 15 days (n = 242), n (%) | 86 (51.8) | 40 (52.6) | 0.905 |
| **Hypertension** | | | |
| Maximum systolic blood pressure (mmHg) (n = 1193) | 166 [150; 187] | 170 [152; 191] | 0.001 |
| **Acute Kidney Failure** | | | |
| Acute kidney failure during ICU Stay (n = 1164), n (%) | 355 (45.1) | 153 (40.6) | 0.145 |
| KDIGO criteria—Stage 1 (n = 1164), n (%) | 196 (24.9) | 88 (23.3) | 0.561 |
| KDIGO criteria—Stage 2 (n = 1164), n (%) | 62 (7.9) | 32 (8.5) | 0.721 |
| KDIGO criteria—Stage 3 (n = 1164), n (%) | 97 (12.3) | 33 (8.8) | 0.070 |
| Urea Ratio (n = 1114), n (%) | 1.02 [1.00; 1.50] | 1.21 [1.00; 1.80] | < .001 |
| **Metabolic Disorders** | | | |
| Hyperglycaemia, n (%) | 228 (26.6) | 167 (42.7) | < .001 |
| Hypoglycaemia, n (%) | 36 (4.1) | 38 (9.7) | < .001 |
| Hypokalemia (n = 1176), n (%) | 285 (36.1) | 138 (35.8) | 0.913 |
| **Digestive adverse events** | | | |
| Digestive bleeding (n = 1127), n (%) | 8 (1.1) | 7 (1.9) | 0.228 |
| Gastric protective agent, n (%) | 462 (54.0) | 275 (70.3) | < .001 |
| Digestive endoscopy, n (%) | 29 (3.4) | 9 (2.3) | 0.301 |
| **Critical illness myopathy** | 20 (4.1) | 15 (6.8) | 0.126 |
| **Delirium** | 13 (2.7) | 4 (1.8) | 0.491 |

Urea Ratio defined by the ratio of maximum urea level during ICU stay over urea level at admission in mmol/L.

ICU: Intensive Care Unit. AECOPD: Acute exacerbation of chronic obstructive pulmonary disease. IMV: Invasive Mechanical Ventilation. LOS: Length of stay.

KDIGO: Kidney Disease Improving Global Outcomes.

Corticosteroid therapy remains recommended for AECOPD in the 2022 GOLD Guidelines [2]. Studies carried out outside ICU units have described a benefit of corticosteroid therapy with regard to various outcomes, such as improvement in $FEV_1$, reduction in treatment failures, and shortening of the length of stay in hospital, but without any effect on mortality [2].

The French recommendations propose to evaluate prescription at the individual patient level [5]. No specific robust recommendations have been made for ICU patients with AECOPD.

Two randomized studies were conducted in ICUs. Alia *et al.* conducted a multicentre study evaluating 83 mechanically ventilated patients randomized to receive methylprednisolone 0.5 mg/kg for 72 hours or placebo, followed by 7-day treatment. Corticosteroids significantly reduced the duration of non-invasive ventilation, length of stay in ICU, and NIV failure. These results might be explained by the unexpectedly high rate of NIV failure in the placebo group (37% in placebo group vs. 0% in corticosteroids group) compared to NIV failure rates close to 20% in literature [16,17]. In contrast to our results, a positive effect of corticosteroids was not found for patients on invasive mechanical ventilation [6]. Likewise, in an open randomized study, where a unique daily dose of 1mg/kg/day of prednisone was prescribed for 10 days in the corticosteroids group, Abroug *et al.* found no effect of corticosteroids on mortality, NIV failures, duration of mechanical ventilation or length of stay in ICU [7]. A meta-analysis of these studies did not reach any robust conclusions regarding the indication for corticosteroid therapy in ICUs [18]. Moreover, these studies excluded important classes of ICU patients, in particular those with AECOPD triggered by pneumonia. Our results showed that in real life practice only 31.4% of patients were treated with corticosteroid therapy reflecting the lack of robust evidence concerning this therapy in the field.

The more impactful result of our study was a reduction in the risk of death or invasive mechanical ventilation at Day 28 (OR = 0.70 [0.49; 0.99], *p = 0.044)* with corticosteroids.

The most severe COPD patients, on home long-term oxygen therapy or long-term non-invasive ventilation, did not benefit from corticosteroid therapy. In patients at an advanced stage of the disease with severe obstruction and emphysema, corticosteroids seem be of minor interest. While, this needs to be confirmed in studies dedicated to these phenotypes, the addition of corticosteroids could simply increase the therapeutic burden.

The results from our large multicentre cohort of patients admitted to ICUs with a good control of confounding factors by IPTW statistical analysis support the routine prescription of corticosteroids for AECOPD in the ICU, regardless of the type of ventilatory support needed or the initial severity scores.

Consistent with the results of our main outcome, we also observed trends for a positive effect of corticosteroids on in-ICU mortality, in-hospital death, and 28-day and 90-day survival rates.

No significant relationship was found between corticosteroid therapy and the prescription of antibiotics except for a trend toward a decrease in antibiotic-free days (AFD) at 10-days for the 395 patients still in the ICU (IRR = 0. 77[0.59; 0.99], *p = 0.046*). However, there was a high rate of prescription of antibiotics at admission, the GOLD guidelines recommending the prescription of antibiotics for all AECOPD when they present severity criteria, and our analysis at 15-days might be flawed by physicians prescribing longer antibiotic therapy in patients with a poor evolution (S5 Fig). Corticosteroid therapy might be indirectly responsible for side effects due to prolonged antibiotic therapies such as antibiotic adverse reactions and the selection of drug-resistant organisms in turn leading to an increase in health related costs [19].

Regarding adverse effects related to corticosteroid therapy, we found no increased incidence of nosocomial infections in patients who received corticosteroid therapy at admission. The difference in maximum systolic pressure was significant but not clinically relevant. Data from the literature promotes short courses with moderate doses of corticosteroids even for ICU patients [3,20,21] minimizing exposure to the drug. In a cohort of 17,239 patients, lower-doses of corticosteroid (methylprednisolone, ≤240 mg/d vs. >240 mg/d) were not associated with a significant reduction in mortality, but with reduced hospital and ICU length of stay, hospital costs, length of IMV, need for insulin therapy, and fungal infections [21]. Overall, in

these studies, critically ill patients admitted for AECOPD were treated with higher doses than the recommended dose for non-ICU patients [20,21]. The question of the optimal corticosteroid dose for ICU patients therefore remains to be explored. In our study, patients in the group treated with corticosteroids had more glycaemic disorders (Table 3) in line with the two consistently reported RCTs [6,18]. This is relevant because, the question of glycaemic control is crucial in an ICU [22] and in the COPD population, hyperglycaemic episodes being associated with poor outcomes and increased rates of NIV failure [23,24].

The strengths of this study are the size of the cohort, the methodology and the quality of data from the Outcomerea database involving 32 ICUs reflecting real life practices.

A limitation was the 43% of patients without known staging of COPD severity. However, a high proportion of patients hospitalized for AECOPD are poorly monitored, or do not even have a known diagnosis of their COPD. Indeed, in patients having mechanical ventilator support for acute hypercapnic respiratory failure in the ICU, nearly 2/3 of patients with COPD were not previously diagnosed [9]. Underdiagnosis and poor monitoring are frequent in COPD patients, explaining the large proportion of patients with unknown COPD severity in our study, reflecting the daily practice concerning AECOPD in ICU. Unassessed confounding factors may exist but the exhaustiveness of our database limits this bias.

Beyond the retrospective nature of the study, the main limitation of our study was the absence of any control of dose and duration of corticosteroid therapy. If the time effect is included in the analysis, evolutions of medical practices over the duration of the study about the management of EACOPD in the ICU could have influenced the outcomes. An ongoing French randomized ICU trial with short corticosteroid therapy, broad inclusion criteria and controlled NIV management will help to confirm or modulate our findings (ClinicalTrials.gov Identifier: NCT04163536). Unfortunately, we do not have data concerning the smoker status (current smokers or former smokers) and concerning the pack years/ smoking index. The level of pack/years index is known associated to the severity of COPD disease but it still debated that current smokers have more AECOPD compare to former smokers as bronchial inflammation persist after smoking cessation [2,25,26]. It does not exist at our knowledge evidence that smoking habits may have impact on the efficiency of systemic corticosteroids therapy in severe AECOPD [2]. Several studies suggest that glucocorticoids may be less efficacious to treat AECOPs in patients with lower level of blood eosinophils [2,27,28]. Unfortunately, we do not have details of the complete blood count.

## Conclusions

In a large cohort of 1,247 patients registered in the OutcomeRea^TM database, corticosteroid treatment on admission to ICU for patients with AECOPD reduced a composite outcome: death or the need for invasive mechanical ventilation at day 28. While patients treated with corticosteroid therapy did not present more nosocomial infections there were more glycaemic disorders. Further studies are needed to better define the phenotypes of responders and the appropriate dosage and duration of corticosteroid treatment.

## Supporting information

**S1 Fig. Survival curves at day 28 according to corticosteroid therapy for AECOPD at admission in ICU.** Effects of corticosteroids on survival analysis (Cox model) for 28-day survival: HR = 0,89 [0.64; 1.24], *p = 0.497*.
(DOCX)

**S2 Fig. Survival curves at day 90 according to corticosteroids therapy for AECOPD at admission in ICU (n = 863).** Effects of corticosteroids in survival analysis (cox model) for 90-day survival: HR = 0.79 [0.6; 1.06], *p = 0.121*.
(DOCX)

**S3 Fig. Survival curves at day 28 according to corticosteroids therapy for AECOPD at admission in ICU (n = 863). Very severe COPD patients.** Effects of corticosteroids in survival analysis (cox model) for 28-day survival for patients with a very severe COPD: HR = 1.21 [0.60; 2.42], *p = 0.598*. COPD: Chronic obstructive pulmonary disease.
(DOCX)

**S4 Fig. Survival curves at day 28 according to corticosteroids therapy for AECOPD at admission in ICU (n = 863). Not very Severe COPD patients or unknown COPD severity.** Effects of corticosteroids in survival analysis (cox model) for 28-day survival for patients with not very severe COPD or unknown COPD Severity: HR = 0.85 [0.58; 1.26], *p = 0.420*. COPD: Chronic obstructive pulmonary disease.
(DOCX)

**S5 Fig. Summary of results for corticosteroids therapy and ventilator-free days. All patients and sub-groups of patients.** VFD at day 28 is express in Median and interquartiles of days, Median [Q1; Q3]. IRR: Incidence Rate Ratio. VFD: Ventilator-free days. IMV: Invasive Mechanical Ventilation. NIV: Non-Invasive Ventilation.
(DOCX)

**S6 Fig. Summary of results for corticosteroids therapy and antibiotic consumption.** ICU: Intensive Care Unit.
(DOCX)

**S7 Fig. Summary of results for corticosteroids therapy regarding the endpoint "Alive and antibiotic-free days" at 5 days, 10 days and 15 days.** AFD is express in Median and interquartiles of days, Median [Q1; Q3]. IRR: Incidence Rate Ratio. AFD: Antibiotic-free days. LOS: Length of stay.
(DOCX)

**S1 Table. Ventilatory support used for patients admitted in ICU for a severe AECOPD.** ICU: Intensive Care Unit. NIV: Non-Invasive Ventilation. IMV: Invasive Mechanical Ventilation.
(DOCX)

**S2 Table. Characteristics of patients regarding the prescription of corticosteroids at ICU admission.** Statistical analysis performed with khi2 Test and Wilcoxon-Mann-Whitney Test. ICU: Intensive Care Unit. AECOPD: Acute exacerbation of chronic obstructive pulmonary disease. BMI: Body Mass Index. SOFA: Sequential Organ Failure Assessment. NIV: Non-Invasive Ventilation. IMV: Invasive Mechanical Ventilation. Pa02: Partial pressure of oxygen. FiO2: Fraction of inspired oxygen. PaCO2: Partial Pressure of Carbon Dioxide.
(DOCX)

**S3 Table. Weight model used to compute IPTW—Corticosteroids treatment for AECOPD at admission in ICU.** Adjustment also performed on centre. ICU: Intensive Care Unit. AECOPD: Acute exacerbation of chronic obstructive pulmonary disease. IPTW: Inverse Probability of Treatment Weighting. BMI: Body Mass Index. SOFA: Sequential Organ Failure Assessment. Pa02: Partial pressure of oxygen. FiO2: Fraction of inspired oxygen. NIV: Non-

Invasive Ventilation. IMV: Invasive Mechanical Ventilation.
(DOCX)

**S4 Table. Double Robust Analysis of the association between prescription of corticosteroids at admission in ICU for AECOPD and primary composite outcome: Death or invasive mechanical ventilation at Day 28.** Adjustment also performed on centre and year. ICU: Intensive Care Unit. AECOPD: Acute exacerbation of chronic obstructive pulmonary disease. IPTW: Inverse Probability of Treatment Weighting. BMI: Body Mass Index. SOFA: Sequential Organ Failure Assessment. Pa02: Partial pressure of oxygen. FiO2: Fraction of inspired oxygen. NIV: Non-Invasive Ventilation. IMV: Invasive Mechanical Ventilation.
(DOCX)

**S5 Table. Double Robust Analysis of the association between prescription of corticosteroids at admission in ICU for AECOPD and in-ICU death.** Adjustment also performed on centre and year. Survival analysis performed with a cox model. ICU: Intensive Care Unit. AECOPD: Acute exacerbation of chronic obstructive pulmonary disease. COPD: Chronic Obstructive Pulmonary Disease. BMI: Body Mass Index. SOFA: Sequential Organ Failure Assessment. Pa02: Partial pressure of oxygen. FiO2: Fraction of inspired oxygen. NIV: Non-Invasive Ventilation. IMV: Invasive Mechanical Ventilation.
(DOCX)

**S6 Table. Double Robust Analysis of the association between prescription of corticosteroids at admission in ICU for AECOPD and in-hospital death.** Adjustment also performed on centre and year. Survival analysis performed with a cox model. ICU: Intensive Care Unit. AECOPD: Acute exacerbation of chronic obstructive pulmonary disease. COPD: Chronic Obstructive Pulmonary Disease. BMI: Body Mass Index. SOFA: Sequential Organ Failure Assessment. Pa02: Partial pressure of oxygen. FiO2: Fraction of inspired oxygen. NIV: Non-Invasive Ventilation. IMV: Invasive Mechanical Ventilation.
(DOCX)

**S1 File. OutcomeRea<sup>TM</sup> database.**
(DOCX)

**S2 File. Precisions on statistical analysis.**
(DOCX)

**S3 File. Side effects definitions.**
(DOCX)

**S4 File. Statistical analysis.** Decrease in the prescription of corticosteroid therapy over the years.
(DOCX)

**S5 File. OUTCOMEREA network.**
(DOCX)

# Acknowledgments

We thank all the medical and research teams members of the OutcomeRea<sup>TM</sup> Network (listed in supporting information).

We thank Alison Foote (Grenoble, France) for writing assistance, technical editing, language editing, and proofreading.

## Author Contributions

**Conceptualization:** Louis Marie Galerneau, Jean-François Timsit.

**Data curation:** Louis Marie Galerneau, Sébastien Bailly, Nicolas Terzi, Stéphane Ruckly, Maïté Garrouste-Orgeas, Yves Cohen, Vivien Hong Tuan Ha, Marc Gainnier, Shidasp Siami, Claire Dupuis, Michael Darmon, Elie Azoulay, Jean-Marie Forel, Florian Sigaud, Christophe Adrie, Dany Goldgran-Toledano, Alexis Ferré, Etienne de Montmollin, Laurent Argaud, Jean Reignier, Jean-Louis Pepin, Jean-François Timsit.

**Formal analysis:** Louis Marie Galerneau.

**Investigation:** Jean-Louis Pepin.

**Methodology:** Louis Marie Galerneau, Sébastien Bailly, Stéphane Ruckly, Jean-François Timsit.

**Supervision:** Nicolas Terzi, Jean-Louis Pepin, Jean-François Timsit.

**Validation:** Stéphane Ruckly, Jean-Louis Pepin, Jean-François Timsit.

**Writing – original draft:** Louis Marie Galerneau.

**Writing – review & editing:** Sébastien Bailly, Nicolas Terzi, Stéphane Ruckly, Maïté Garrouste-Orgeas, Yves Cohen, Vivien Hong Tuan Ha, Marc Gainnier, Shidasp Siami, Claire Dupuis, Michael Darmon, Elie Azoulay, Jean-Marie Forel, Florian Sigaud, Christophe Adrie, Dany Goldgran-Toledano, Alexis Ferré, Etienne de Montmollin, Laurent Argaud, Jean Reignier, Jean-Louis Pepin, Jean-François Timsit.

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
