## [Decision Letter · Decision Letter 0]

23 Nov 2022

PONE-D-22-27411Corticosteroids for severe acute exacerbations of chronic obstructive pulmonary disease in intensive care: from the French OUTCOMEREA cohort.PLOS ONE

Dear Dr. Galerneau,

Thank you for submitting your manuscript to PLOS ONE. After careful consideration, we feel that it has merit but does not fully meet PLOS ONE’s publication criteria as it currently stands. Therefore, we invite you to submit a revised version of the manuscript that addresses the points raised during the review process.

ACADEMIC EDITOR:Dear Author,

Your work has been very interesting. But we need some revisions. Hopefully, you will provide these fixes as soon as possible. We are waiting for your reply. We expect you to carefully respond to reviewer suggestions and make any necessary corrections.

Best Regards

We look forward to receiving your revised manuscript.

Kind regards,

Bora Çekmen

Academic Editor

PLOS ONE

Journal Requirements:

"Nicolas Terzi is supported by Pfizer for attending meetings and/or travel. The other authors declare that they have no competing interests."

Reviewers' comments:

Reviewer's Responses to Questions

**Comments to the Author**

1. Is the manuscript technically sound, and do the data support the conclusions?

Reviewer #1: Partly

Reviewer #2: Yes

2. Has the statistical analysis been performed appropriately and rigorously? 

Reviewer #1: Yes

Reviewer #2: Yes

3. Have the authors made all data underlying the findings in their manuscript fully available?

Reviewer #1: Yes

Reviewer #2: Yes

4. Is the manuscript presented in an intelligible fashion and written in standard English?

Reviewer #1: Yes

Reviewer #2: Yes

5. Review Comments to the Author

Reviewer #1: Though the study size is large , I find that there are some limitations to this study which I would like the authors to comment upon.

1) How was the diagnosis of COPD made in these patients?It looks like a clinical diagnosis because many of the patients did not have a OFT recorded at some point of time in their life.

2) What linings have the authors taken into consideration for defining an AE of CoOD?

3)How many of these patients were current smokers? What were the pack years/ smoking index in the patients? Do you think these could impacted the outcome?

4) How many of the patients had peripheral blood eosinophilia? Was the response to steroids in these people different?

5) Were people with structural lung disease like bronchiectasis, post TB fibrosis included in the study?

6) The authors have mentioned about hyperglycaemic episodes, sepsis status, hypertension, GI bleed etc . Did any of these patients develop myopathy or any psychiatric symptoms?

7) This study could not come to a conclusion regarding the dosage or duration of AECOPD that could be beneficial and this has already been pointed out by the authors.

Thank you.

Reviewer #2: The manuscript deals with an interesting topic, even if the current knowledge should be clearly enough in this field. In fact, the most recent release of the International guidelines on COPD (2023) state that in patients with severe exacerbations systemic corticosteroids treatment is key point for the management with evidence A (Global Initiative for Chronic Obstructive Pulmonary Disease, 2023 Gold Report, read @ https://goldcopd.org/2023-gold-report-2/). Nevertheless, timing to ICU admission may vary from a hospital to another, and the advantage to use corticosteroids at ICU admission may not be established elsewhere. thus, it should be of interest to know which was the timing of ICU admission with regard to hospital/ER admission, and to include this into the multivariate analysis. Another lacking point is the value of pH/PaCO2 at ICU admission. The knowledge of pH is important to characterize the severity of COPD exacerbation and its prognosis, but also the risk of failure of NIV (See and cite Confalonieri M, et al. Eur Respir J 2005; 25: 348-355). The Authors repeatedly state that acute exacerbation of chronic obstructive pulmonary disease (AECOPD) is one of the most frequent causes intensive care unit (ICU) admission, but I note that only 7% of the ICU admissions in the OUTCOMEREA cohort were caused by AECOPD. Furthermore, the common experience of most ICUs in western Countries don't include AECOPD among the most frequent reasons of ICU admission (e.g. see https://www.ottawahospital.on.ca/en/clinical-services/my-icu-the-intensive-care-unit/icu-patients/icu-medical-conditions/ and others).

6. PLOS authors have the option to publish the peer review history of their article (what does this mean?). If published, this will include your full peer review and any attached files.

Reviewer #1: No

Reviewer #2: No

---

## [Author Response · Author response to Decision Letter 0]

2 Feb 2023

January 27th, 2023

Emily Chenette, 

Editor in Chief

PLOS ONE

Dear Dr. Chenette,

Thank you for giving us the opportunity to submit a revised version of our manuscript entitled “Corticosteroids for severe acute exacerbations of COPD in intensive care: from the French OUTCOMEREA cohort” (PONE-D-22-27411). We appreciate the time and effort that you and the reviewers have dedicated to providing your valuable feedback on our work. We are grateful to the reviewers for their insightful comments on our article. We have carefully addressed all the comments they made. We provide a point-by-point response to the reviewers’ comments and concerns. We are resubmitting a “unmarked” and a “with track changes” version (modifications highlighted in red) of the manuscript. 

We hope the revised manuscript will better meet the PLOS ONE’s high standards for publication. 

We look forward to hearing from you in the near future.

Yours sincerely,

Louis-Marie Galerneau (Corresponding Author)

 

Associate Editor: 

C1- Please ensure that your manuscript meets PLOS ONE's style requirements, including those for file naming.

R1- The manuscript was modified to meet PLOS ONE's style requirements, including for file naming.

C2- Please provide additional details regarding participant consent. In the ethics statement in the Methods and online submission information, please ensure that you have specified (1) whether consent was informed and (2) what type you obtained (for instance, written or verbal, and if verbal, how it was documented and witnessed). If your study included minors, state whether you obtained consent from parents or guardians. If the need for consent was waived by the ethics committee, please include this information.

R2- This study did not require individual patient consent because conducted from a previously approved database by our institutional review board of Clermont-Ferrand, France (IRB no. 5891), which waived the need for signed informed consent of the participants, in accordance with French legislation on non-interventional studies. The study did not modify patients' management and the data were anonymously collected and our study not included minors.

In the materials and methods, we have replaced: 

 “The institutional review board of Clermont-Ferrand University Hospital, France (IRB no. 5891) approved the study (no. 2007–2016)." 

by 

"This study did not require individual patient consent because conducted from a previously approved database by our institutional review board of Clermont-Ferrand, France (IRB no. 5891; Ref: 2007–16), which waived the need for signed informed consent of the participants, in accordance with French legislation on non-interventional studies. ".

Several studies analysing data from the OutcomeReaTM database has already been published in PLOS ONE [1–3] .

R2- References

1. Papin G, Bailly S, Dupuis C, Ruckly S, Gainnier M, Argaud L, et al. Clinical and biological clusters of sepsis patients using hierarchical clustering. PloS One. 2021;16: e0252793. doi:10.1371/journal.pone.0252793

2. Dupuis C, de Montmollin E, Buetti N, Goldgran-Toledano D, Reignier J, Schwebel C, et al. Impact of early corticosteroids on 60-day mortality in critically ill patients with COVID-19: A multicenter cohort study of the OUTCOMEREA network. PloS One. 2021;16: e0255644. doi:10.1371/journal.pone.0255644

3. Ibn Saied W, Souweine B, Garrouste-Orgeas M, Ruckly S, Darmon M, Bailly S, et al. Respective impact of implementation of prevention strategies, colonization with multiresistant bacteria and antimicrobial use on the risk of early- and late-onset VAP: An analysis of the OUTCOMEREA network. PloS One. 2017;12: e0187791. doi:10.1371/journal.pone.0187791

 

C3- Thank you for stating the following financial disclosure: 

R3- 

a) We specified in the funding section of the declarations: “S.B. and J.-L.P. are supported by the French National Research Agency in the framework of the ‘Investissements d’avenir’ program (ANR-15-IDEX-02) and Grenoble Alpes University Foundation Chairs of excellence: “e-health and integrated care and trajectories medicine and MIAI artificial intelligence” This work has been partially supported by MIAI @ Grenoble Alpes (ANR-19-P3IA-0003).”

b) We added in the funding section of the declarations: “The funders had no role in study design, data collection and analysis, decision to publish, or preparation of the manuscript.”

c) None authors received a salary from any of our funders.

d) We added in the funding section of the declarations: “The authors received no specific funding for this work."

 

C4- Thank you for stating the following in the Competing Interests section: 

"Nicolas Terzi is supported by Pfizer for attending meetings and/or travel. The other authors declare that they have no competing interests."

R4- We have added in the competing interests section of the declarations: "This does not alter our adherence to PLOS ONE policies on sharing data and materials.”

 

Reviewer #1:

Comments to the Author

C1- How was the diagnosis of COPD made in these patients? It looks like a clinical diagnosis because many of the patients did not have an OFT recorded at some point of time in their life.

R1- Indeed, the diagnosis was proved by pulmonary function tests in 290 patients with pulmonary function tests results registered in the database. For the others patients, the diagnosis was known by medical history and comorbidities registered in the database. These data may be communicated by the patients, the relatives of the patients during ICU stay, or documented in their medical files.

If there was a strong clinical suspicion of COPD, the ICU practitioner could register the presence of COPD in the database. This situation is frequent in daily practice. Indeed, a large part of patients admitted in ICU for severe acute respiratory failure, especially with hypercapnic respiratory failure, have an undiagnosed COPD [4].

This paragraph and reference 4 have been included in the revised version of the manuscript in the Materials and methods - Study design and study population section.

R1- References

4. Adler D, Pépin J-L, Dupuis-Lozeron E, Espa-Cervena K, Merlet-Violet R, Muller H, et al. Comorbidities and Subgroups of Patients Surviving Severe Acute Hypercapnic Respiratory Failure in the ICU. Am J Respir Crit Care Med. 2016. doi:10.1164/rccm.201608-1666OC 

C2 - What linings have the authors taken into consideration for defining an AE of COPD?

R2-We thank the reviewer for their suggestions as how to improve the clarity of the methods presentation.

To clarify this point, we have included in the materials and methods a new section the following sentences to address this comment: 

" We included adults admitted to one of the 32 ICUs participating in OutcomeReaTM with a diagnosis of acute exacerbation of COPD. The diagnosis of EACOPD was define by a main diagnosis of exacerbation of COPD registered in the database or by a main diagnosis registered of acute respiratory failure with a medical history of COPD registered in the database. Patients included was admitted in ICU between January 1, 1997 and December, 31 2018.". 

 

C3 - How many of these patients were current smokers? What were the pack years/ smoking index in the patients? Do you think these could impacted the outcome?

R3 - Unfortunately, we do not have data concerning the smoker status, current smokers or former smokers and concerning the pack years/ smoking index. It should be a data that we will collect in the future in the OUTCOMEREATM database to further improve the quality of the data.

Concerning the impact of this data about smoking on the outcomes, the level of pack/years index is known associated to the severity of COPD disease but it still debated that current smokers have more AECOPD compare to former smokers as bronchial inflammation persist after smoking cessation [5–7]. It does not exist at our knowledge evidence that smoking habits may have impact on the efficiency of systemic corticosteroids therapy in severe AECOPD [7].

We have acknowledged as a limitation the fact that we have no data regarding smoking status and included in the Discussion section the sentence: 

“Unfortunately, we do not have data concerning the smoker status (current smokers or former smokers) and concerning the pack years/ smoking index. The level of pack/years index is known associated to the severity of COPD disease but it still debated that current smokers have more AECOPD compare to former smokers as bronchial inflammation persist after smoking cessation [2,25,26]. It does not exist at our knowledge evidence that smoking habits may have impact on the efficiency of systemic corticosteroids therapy in severe AECOPD [2].”

R3- References

5. Adibi A, Sin DD, Safari A, Johnson KM, Aaron SD, FitzGerald JM, et al. The Acute COPD Exacerbation Prediction Tool (ACCEPT): a modelling study. Lancet Respir Med. 2020;8: 1013–1021. doi:10.1016/S2213-2600(19)30397-2

6. Hurst JR, Vestbo J, Anzueto A, Locantore N, Müllerova H, Tal-Singer R, et al. Susceptibility to exacerbation in chronic obstructive pulmonary disease. N Engl J Med. 2010;363: 1128–1138. doi:10.1056/NEJMoa0909883

7. Global Initiative for Chronic Obstructive Lung Disease (GOLD). Global Strategy for the Diagnosis, Management and Prevention of COPD. 2022 Report. www.goldcopd.org.  

C4 - How many of the patients had peripheral blood eosinophilia? Was the response to steroids in these people different?

R4 - Thank you for your relevant comment. Several studies suggest that glucocorticoids may be less efficacious to treat AECOPs in patients with lower level of blood eosinophils [7–9].

Unfortunately, we do not have details of the complete blood count. We have data about daily leucocytes count but without precision regarding the specific level of blood eosinophils. 

We have acknowledged as a limitation the fact that we have no data regarding eosinophilia and included in the Discussion section references 7 to 9 and the sentence: 

 “Several studies suggest that glucocorticoids may be less efficacious to treat AECOPs in patients with lower level of blood eosinophils [2,27,28]. Unfortunately, we do not have details of the complete blood count.”

R4- References

 7. Global Initiative for Chronic Obstructive Lung Disease (GOLD). Global Strategy for the Diagnosis, Management and Prevention of COPD. 2022 Report. www.goldcopd.org. 

8. Sivapalan P, Lapperre TS, Janner J, Laub RR, Moberg M, Bech CS, et al. Eosinophil-guided corticosteroid therapy in patients admitted to hospital with COPD exacerbation (CORTICO-COP): a multicentre, randomised, controlled, open-label, non-inferiority trial. Lancet Respir Med. 2019;7: 699–709. doi:10.1016/S2213-2600(19)30176-6

9. Bafadhel M, McKenna S, Terry S, Mistry V, Pancholi M, Venge P, et al. Blood eosinophils to direct corticosteroid treatment of exacerbations of chronic obstructive pulmonary disease: a randomized placebo-controlled trial. Am J Respir Crit Care Med. 2012;186: 48–55. doi:10.1164/rccm.201108-1553OC 

C5 - Were people with structural lung disease like bronchiectasis, post TB fibrosis included in the study?

R5 - Concerning patients with pulmonary fibrosis, they have been excluded of the analysis (Figure 1), regardless of the type of the pulmonary fibrosis. This included patients with post TB fibrosis. 

Concerning bronchiectasis, only 4 patients had a diagnosis of bronchiectasis known and reported.  

C6 - The authors have mentioned about hyperglycaemic episodes, sepsis status, hypertension, GI bleed etc. Did any of these patients develop myopathy or any psychiatric symptoms?

R6 - We thank the reviewer for this very relevant comment. We have now included data about critical illness myopathy and delirium in the new table 3.

We found 20 patients (4.1%) with Critical illness myopathy in the group without corticosteroids and 15 patients (6.8%) in the group with corticosteroids, without differences between the groups (p=0.126). 

Concerning delirium, we found 13 patients (2.7%) with diagnosis of delirium in no corticosteroids group and 4 patients (1.8%) (p=0.491).

 

C7 - This study could not come to a conclusion regarding the dosage or duration of AECOPD that could be beneficial and this has already been pointed out by the authors.

R7 – We fully agree with your comment. Dosage of corticosteroids should be a data that we will collected in the future in the OUTCOMEREATM database.

Specific studies in ICU about dosage and duration of corticosteroids prescription for AECOPD are needed.

 

Reviewer #2: 

C1 - The manuscript deals with an interesting topic, even if the current knowledge should be clearly enough in this field. In fact, the most recent release of the International guidelines on COPD (2023) state that in patients with severe exacerbations systemic corticosteroids treatment is key point for the management with evidence A (Global Initiative for Chronic Obstructive Pulmonary Disease, 2023 Gold Report, read @ https://goldcopd.org/2023-gold-report-2/). Nevertheless, timing to ICU admission may vary from a hospital to another, and the advantage to use corticosteroids at ICU admission may not be established elsewhere. 

R1 - We agree and thank the reviewer for this comment.

 

C2 - Thus, it should be of interest to know which was the timing of ICU admission with regard to hospital/ER admission, and to include this into the multivariate analysis.

R2 - Thank you for this very important comment. 

So, we added data about the timing of ICU admission in the new Table 2 presenting characteristics of patients. 1030 (82.6%) patients were directly admitted in ICU or in less than 24h after hospital admission, 115 (9.2%) patients were admitted in ICU between 24h and 7 days after hospital admission and 102 (8.2%) patients were admitted in ICU more than 7 days after hospital admission.

We have added in the results section the following sentence: "The major part of patients was directly admitted in ICU or during the first 24 hours after hospital admission (1030, 82,6%)."

The data about the timing of ICU admission had been added in the new S2 Table presenting characteristics of patients regarding the prescription of corticosteroids at ICU admission. 

We finally added the timing of ICU admission in the weight model used to compute IPTW (new S3 Table) and in the multivariate weighted analysis (new S4, S5, S6 Tables).

These modifications slightly changed the numerical values without any modification in our interpretation and conclusions.

We following changes have been done in the revised version: 

“In a multivariable analysis, after IPTW weighting (double robust analysis) (weight model used resumed in Table E3), corticosteroid administration at ICU admission in ICU significantly improved the principal composite outcome (OR = 0.693 [0.489; 0.98], p=0.038) (Table E4, Figure 2). “

Replaced by 

“In a multivariable analysis, after IPTW weighting (double robust analysis) (weight model used resumed in S3 Table), corticosteroid administration at ICU admission in ICU significantly improved the principal composite outcome (OR = 0.70 [0.49; 0.99], p=0.044) (S4 Table, Fig 2).”

“The subgroup analyses on the principal composite endpoint are summarized in Fig 2. For the subgroup of patients with very severe COPD, the protective effect of corticosteroid therapy on the primary composite outcome was lost (OR = 1.12 [0.53; 2.37], p=0. 770).”

Replaced by 

“The subgroup analyses on the principal composite endpoint are summarized in Fig 2. For the subgroup of patients with very severe COPD, the protective effect of corticosteroid therapy on the primary composite outcome was lost (OR = 1.12 [0.53; 2.37], p=0. 770).”

“Fig 2. Summary of results for the primary outcome (death or invasive mechanical ventilation at day 28). Heterogeneity for COPD severity subgroups: Chi2 = 3.50, df = 2, p = 0.32. Heterogeneity for ventilatory support subgroups: Chi2 = 6.36, df = 2, p = 0.10. Heterogeneity for SAPS II subgroups: Chi2 = 5,26, df = 1, p = 0.15.”

Replaced by 

“Figure 2. Summary of results for the primary outcome (death or invasive mechanical ventilation at day 28).

Heterogeneity for COPD severity subgroups: Chi2 = 2.29, df = 2, p = 0.51

Heterogeneity for ventilatory support subgroups: Chi2 = 1.45, df = 2, p = 0.69

Heterogeneity for SAPS II subgroups: Chi2 = 0.49, df = 1, p = 0.92”

We not presented here the modifications for the secondary outcomes, but they are marked in the new version of the manuscript, and are minor modifications.

Therefore, we also slightly modified the Figure 2, in the Figures S5, S6, S7 and in the legends of the Figures S1, S2, S3, S4 

C3 - Another lacking point is the value of pH/PaCO2 at ICU admission. The knowledge of pH is important to characterize the severity of COPD exacerbation and its prognosis, but also the risk of failure of NIV (See and cite Confalonieri M, et al. Eur Respir J 2005; 25: 348-355).

R3 - The values of arterial blood gas at ICU admission are presented in the Table 2, including value of pH and PaCO2. The pH value in presented by the median of pH and by severity categories defined by the cut-off values presented in Confalonieri M, et al. Eur Respir J 2005; 25: 348-355.

The value of pH is used in the weight model used to compute IPTW (Table S3) and in the weighted multivariate analyses. 

The value of PaCO2 is not used in these analyses because pH and PaCO2 presented collinearity in this population.

We cited Confalonieri M, et al. Eur Respir J 2005; 25: 348-355. in the article and we modified in the results section as follow:

"The clinical characteristics of the study population at ICU admission are shown in Table 1 and the characteristics of COPD exacerbations are given in Table 2."

Replaced by

"The clinical characteristics of the study population at ICU admission are shown in Table 1. The characteristics of COPD exacerbations and arterial blood gas result which is a prognosis factor [11] are given in Table 2."

 

C4 - The Authors repeatedly state that acute exacerbation of chronic obstructive pulmonary disease (AECOPD) is one of the most frequent causes intensive care unit (ICU) admission, but I note that only 7% of the ICU admissions in the OUTCOMEREA cohort were caused by AECOPD. Furthermore, the common experience of most ICUs in western Countries don't include AECOPD among the most frequent reasons of ICU admission (e.g. see https://www.ottawahospital.on.ca/en/clinical-services/my-icu-the-intensive-care-unit/icu-patients/icu-medical-conditions/ and others).

R4 - We agree and thank the reviewer for these comments. We have tone down our statement regarding the high prevalence of AECOPD in ICU.

We modified in the abstract section:

“Acute exacerbation of chronic obstructive pulmonary disease (AECOPD) is one of the most frequent causes intensive care unit (ICU) admission. “

by 

“Acute exacerbation of chronic obstructive pulmonary disease (AECOPD) is a frequent cause of intensive care unit (ICU) admission.”

We modified in the introduction section:

“Severe acute exacerbation of chronic obstructive pulmonary disease (AECOPD) is a frequent cause of admission to an intensive care unit (ICU) and may require non-invasive or invasive ventilation support [1].”

by 

“Severe acute exacerbation of chronic obstructive pulmonary disease (AECOPD) is a frequent cause of admission to an intensive care unit (ICU) and may require non-invasive or invasive ventilation support [1].”

We modified in the introduction section:

“This is a question of major importance owing to the high prevalence of AECOPD among ICU admissions and the potential high impact in terms of medical and/or economic benefits.”

by 

“This is a question of major importance owing the potential high impact in terms of medical and/or economic benefits.”

---

## [Decision Letter · Decision Letter 1]

4 Apr 2023

Corticosteroids for severe acute exacerbations of chronic obstructive pulmonary disease in intensive care: from the French OUTCOMEREA cohort.

PONE-D-22-27411R1

Dear Dr. Galerneau,

We’re pleased to inform you that your manuscript has been judged scientifically suitable for publication and will be formally accepted for publication once it meets all outstanding technical requirements.

Kind regards,

Samuele Ceruti

Academic Editor

PLOS ONE

Additional Editor Comments (optional):

Reviewers' comments:

Reviewer's Responses to Questions

**Comments to the Author**

1. If the authors have adequately addressed your comments raised in a previous round of review and you feel that this manuscript is now acceptable for publication, you may indicate that here to bypass the “Comments to the Author” section, enter your conflict of interest statement in the “Confidential to Editor” section, and submit your "Accept" recommendation.

Reviewer #1: All comments have been addressed

Reviewer #2: All comments have been addressed

2. Is the manuscript technically sound, and do the data support the conclusions?

Reviewer #1: Yes

Reviewer #2: Yes

3. Has the statistical analysis been performed appropriately and rigorously? 

Reviewer #1: Yes

Reviewer #2: Yes

4. Have the authors made all data underlying the findings in their manuscript fully available?

Reviewer #1: Yes

Reviewer #2: Yes

5. Is the manuscript presented in an intelligible fashion and written in standard English?

Reviewer #1: Yes

Reviewer #2: Yes

6. Review Comments to the Author

Reviewer #1: All comments have been addressed satisfactorily and the article may be considered for publication if found to meet the standards of the journal

Reviewer #2: The Authors fullfilled the suggestions by the reviewers. The paper is now more readible and comprehensive.

7. PLOS authors have the option to publish the peer review history of their article (what does this mean?). If published, this will include your full peer review and any attached files.

Reviewer #1: No

Reviewer #2: No

---

## [Editor Report · Acceptance letter]

11 Apr 2023

PONE-D-22-27411R1 

Corticosteroids for severe acute exacerbations of chronic obstructive pulmonary disease in intensive care: from the French OUTCOMEREA cohort. 

Dear Dr. Galerneau:

I'm pleased to inform you that your manuscript has been deemed suitable for publication in PLOS ONE. Congratulations! Your manuscript is now with our production department. 

Kind regards, 

on behalf of

Dr. Samuele Ceruti 

Academic Editor

PLOS ONE